The Italian version of the extended Barcelona Music Reward Questionnaire (eBMRQ): a validation study and association with age, gender, and musicianship

Carraturo Giulio giulio.carraturo@studio.unibo.it 1 2
Ferreri Laura 3
Cardona Gemma 4
Lorenzo-Seva Urbano 5
Rodriguez-Fornells Antoni 4
Brattico Elvira 1 2
1 Department of Education, Psychology, Communication, University of Bari Aldo Moro , Bari , Italy
2 Center for Music in the Brain, Department of Clinical Medicine, Aarhus University and The Royal Academy of Music , Aarhus , Denmark
3 Department of Brain & Behavioral Science, University of Pavia , Pavia , Italy
4 Department of Cognition, Development and Educational Psychology, Institute of Neurosciences, University of Barcelona , Barcelona , Spain
5 Faculty of Sciences of Education and Psychology, Universitat Rovira i Virgili , Tarragona , Spain
Chen Chong
Electronic publication date: 2025 Feb 28
Publication date: 2025
Volume: 13
Electronic Location ID: e18403
Received 2023 Nov 24; Accepted 2024 Oct 4
Copyright: ©2025 Carraturo et al.
Copyright year: 2025
Copyright holder: Carraturo et al.
License: This is an open access article distributed under the terms of the Creative Commons Attribution License, which permits unrestricted use, distribution, reproduction and adaptation in any medium and for any purpose provided that it is properly attributed. For attribution, the original author(s), title, publication source (PeerJ) and either DOI or URL of the article must be cited.
License URL: https://creativecommons.org/licenses/by/4.0/

Keywords: Music reward, Music perception, Individual differences, Musical expertise, Musical pleasure

Funding: MUR PNRR PE00000015, AGE-IT, CUP: H33C22000680006 Extended Partnership initiative on Neuroscience and neuropharmacology PE00000006, MNESYS, CUP: H93C22000660006 The Center for Music in the Brain is funded by the Danish National Research Foundation DNRF117 This research was supported by EU funding within the MUR PNRR, a novel public-private alliance to generate socioeconomic, biomedical and technological solutions for an inclusive Italian ageing society (project n. PE00000015, AGE-IT, CUP: H33C22000680006). Also, this research was supported by EU funding within the MUR PNRR Extended Partnership initiative on Neuroscience and neuropharmacology (Project no. PE00000006, MNESYS, CUP: H93C22000660006). The Center for Music in the Brain is funded by the Danish National Research Foundation (DNRF117). The funders had no role in study design, data collection and analysis, decision to publish, or preparation of the manuscript.

==============================
Background

Music is a primary source of pleasure for humans. Nevertheless, there is large interindividual variability in how individuals experience and derive pleasure from music and music-related activities. With this study we propose and validate the Italian version of the extended Barcelona Music Reward Questionnaire (eBMRQ), the most in-depth and comprehensive tool for investigating the diverse characterization of individual sensitivity to pleasure in music. In addition, we aim to investigate eBMRQ scores as a function of age, gender, and musicianship across Italian population.

Methods

For the validation process of the Italian eBMRQ, we first conducted forward and backward translation from the original English eBMRQ version. The new Italian version was then administered to 1,012 participants who were fluent in Italian from the north and the south of Italy through online surveys (age range 18–86 years old; M = 34.9, SD = 16.9, females 74%). Unrestricted confirmatory analysis was computed for both six-factor and single-factor models. The effect of gender, age, and musicianship on eBMRQ scores was analyzed through analysis of variance (ANOVA).

Results

The quality assessment of the factor solution indicated that the Italian eBMRQ demonstrated acceptable quality and reliability, making it a valid tool for assessing sensitivity to music reward. All factors were significantly correlated with each other, in line with previous adaptations of the BMRQ. Our findings indicate that females reported higher music reward sensitivity compared to males, except for Social Reward subscale. Moreover, individual reward sensitivity was significantly higher among musicians and amateurs compared to non-musicians, although this trend did not emerge for Sensory-motor and Mood Regulation subscales. Also, overall musical reward sensitivity was negatively associated with age.

Conclusions

The results obtained suggest the feasibility of applying the Italian version of eBMRQ as a reliable tool in the field of affective and clinical music-related research. Furthermore, the significant associations we have highlighted between eBMRQ scores, gender, age, and musicianship contribute to emphasizing the significant impact of individual factors on music reward sensitivity.

Introduction

Music is an art form that has a relevant emotional impact in people’s life, as a wide spectrum of emotions may derive from it, including joy, sadness, excitement, and nostalgia (Reybrouck, Vuust & Brattico, 2021; Sachs, Damasio & Habibi, 2015; Vuilleumier & Trost, 2015). Among these, pleasure plays a fundamental role in shaping our musical experiences (Brattico & Pearce, 2013). Pleasant music activates brain regions associated with reward, motivation, and emotion, such as the nucleus accumbens, amygdala, and ventral tegmental area (Blood & Zatorre, 2001), as well as pleasure derived from musical experiences plays a crucial role in strengthening social connections and community cohesion (Tarr, Launay & Dunbar, 2014). According to Juslin & Västfjäll (2008), the experience of pleasure in music arises from the interaction between the listener’s expectations, emotional responses, and the perceived qualities of the music itself. The investigation into music-induced pleasure has pointed out that the rewarding response elicited by music share commonalities with the gratification derived from fulfilling fundamental survival-related needs (e.g., food) (for a review see Mas-Herrero et al., 2021). Indeed, neuroimaging studies consistently highlighted the engagement of brain regions associated with dopamine-mediated mesocorticolimbic pathways in the pleasurable experience of music (Blood & Zatorre, 2001; Brattico et al., 2016; Ferreri et al., 2019; Ferreri & Rodriguez-Fornells, 2022; Martínez-Molina et al., 2016). Considering that the release of dopamine is associated with motivation and reinforcement processes (Salimpoor et al., 2011), pleasure might also act as a reinforcer, motivating and promoting continued engagement with music. However, it has been suggested that enjoying music appears to be more complex than experiences evoked by primary rewards. Indeed, the pleasant experience of music seems to depend on the modulation of several neural mechanisms, not only those related to emotion but also to attention and memory (Ferreri et al., 2019; Mas-Herrero et al., 2021). According to these studies, attention enhances engagement and emotional anticipation, while memory retrieval adds depth and personal significance to the musical experience. The integration of these neural mechanisms creates a comprehensive and rewarding musical experience. The reasons why music is considered one of the greatest human pleasures are manifold. Music can facilitate the release of emotions, enhance enjoyment, provide comfort, and alleviate stress, making it a pivotal tool for emotional regulation processes (Ferreri et al., 2021; Van Goethem & Sloboda, 2011). Indeed, the ability of music to regulate mood has been identified among the most essential reasons for engaging in musical activities on a daily basis (for a review see Saarikallio & Erkkilä, 2007). For instance, listening to sad music may be beneficial to sort out negative feelings and thoughts, resulting in more positive reported feelings (Eerola et al., 2018; Van den Tol & Edwards, 2015).

Another relevant aspect to consider is music potential to act as a social facilitator, fostering human social activities and promoting individuals to bond into groups (Keller, Novembre & Hove, 2014; Nummenmaa, Putkinen & Sams, 2021; Savage et al., 2021). In addition, several individual factors may influence the experience of pleasure in music, such as empathy (Carraturo et al., 2022; Wallmark, Deblieck & Iacoboni, 2018), musical expertise (Brattico et al., 2016), and even fluid intelligence (Bonetti & Costa, 2016), as well as cultural and situational context (Curzel et al., 2024; Curzel et al., 2023; North & Hargreaves, 2000; Soley & Hannon, 2010). This results in a considerable interindividual variability in the experience of musical pleasure, and an inherent complexity in objectively measuring this construct. In the last decade, however, the implementation of the Barcelona Music Reward Questionnaire (BMRQ) (Mas-Herrero et al., 2013) has enabled a profound and comprehensive exploration of how people experience pleasure and reward from music-related activities. The BMRQ, which the original authors validated in both Spanish and English, allows for a fine-grained assessment of individuals’ reward sensitivity to musical experiences decomposed into five factors: Musical Seeking, Emotion Evocation, Mood Regulation, Social Reward, and Sensory-Motor. Musical Seeking refers to individuals’ tendency to engage in music-related activities, such as attending concerts, or to seek additional information about the music listened (including details about composers); Emotion Evocation relates to music’s capacity to induce relevant emotional responses in individuals; Mood Regulation scale provides an evaluation of how music is employed to regulate mood; the ability of music to promote and enhance social interaction is assessed by Social Reward scale; and finally, Sensory-Motor scale assesses the capacity of music to intuitively induce body movements synchronized to a rhythm’s beat. Studies employing the BMRQ have pointed out several significant findings. For instance, individual sensitivity to musical reward is higher in females compared to males (Fuentes-Sánchez et al., 2023; Mas-Herrero et al., 2013). Also, music-related pleasure tends to decline with age (Mas-Herrero et al., 2013), as further corroborated in a recent cross-sectional study on adult population (Belfi et al., 2022). As regards the effect of musical expertise on music reward sensitivity, prior evidence revealed that musicians score higher than non-musicians on overall BMRQ (Fuentes-Sánchez et al., 2023; Hernández et al., 2019; Mas-Herrero et al., 2013). Recently, Cardona et al. (2022) proposed and validated an extended version of BMRQ (eBMRQ), adding to the five main factors a sixth facet assessing individuals’ experiences of absorption in music listening (Absorption in Music). The additional four items composing the Absorption in Music scale were taken from the Absorption in Music scale (Sandstrom & Russo, 2013), developed in English. The concept of absorption can be described as a willingness to be deeply drawn in by sensory stimuli, experiencing immersion without distraction (Sandstrom & Russo, 2013). It can be considered similar to the concept of flow (Nakamura & Csikszentmihalyi, 2002), hence a profound emotional involvement and connection that a person may experience while listening to or performing music. In other words, the Absorption in Music scale addresses the degree of immersion and how deeply an individual engages in the musical experience, not only on the intensity and quality of emotions elicited by music (as assessed by Emotional Evocation scale). High scores in trait absorption lead to more intense emotional reactions in response to music (Kreutz et al., 2008), as well as a positive relationship has been pointed out between trait absorption and the enjoyment of evoked negative emotions in response to music (Garrido & Schubert, 2011). Absorption trait is also associated with listening habits, musical preferences, and overall music experience, as it can affect amount of time spent listening to music every day, the importance of music, and knowledge about preferred music or artists (Sandstrom & Russo, 2013; Wild, Kuiken & Schopflocher, 1995). As such, the eBMRQ aims to serve as an even more in-depth and comprehensive tool for investigating the diverse characterization of individual sensitivity to pleasure in music. The main aim of this study is to develop and validate an Italian version of the extended Barcelona Music Reward Questionnaire (eBMRQ), and to assess its psychometric properties, including its factor structure. A further aim of this study is to investigate eBMRQ scores as a function of age, gender, and musicianship across Italian population.

Materials & Methods

Instruments

The original extended version of Barcelona Music Reward Questionnaire (eBMRQ), here used in accordance with a Creative Commons Attribution 4.0 International License (CC BY 4.0), provides a measure of sensitivity to musical reward by describing situations that individuals could experience in their daily life. The questionnaire consists of 24 items divided into six facets (Musical Seeking (MS), Emotion Evocation (EE), Mood Regulation (MR), Social Reward (SR), Sensory-Motor (SM), and Absorption in Music (AM). Participants are requested to indicate the level of agreement with each statement by using a five-point scale from fully disagree (1) to fully agree (5). ORION reliability of the six subscales ranged from 0.836 to 0.932, whereas the ORION reliability of the overall dimension was 0.952. Please note that, while Cronbach’s alpha a measure of internal consistency, that is, how closely related a set of items are as a group, ORION is a reliability index of factor scores that it includes in the estimate of the reliability the inter-factor correlation values (Ferrando & Lorenzo-Seva, 2016). The Italian version of the eBMRQ was developed following the forward–backwards translation method (Brislin, 1970). First, the original English version of the eBMRQ was independently translated into Italian by two Italian native speakers fluent in English and Spanish. The original Spanish questionnaire served as a point of reference to clarify some wordings. The Italian version obtained was then back-translated into English by an English native speaker fluent in Italian who was blind to the original version. For the translation, greater emphasis was placed on capturing meaning rather than literal translation (see Appendix A in the Supplementary Files for the Italian version of the eBMRQ).

Participants

A total of 1,012 adult participants (74% women, 34.9 ± 16.9 years old) fluent in Italian took part in the study. To enable greater generalizability of the results to the Italian population, participants were recruited from Bari (n = 452) in Southern Italy, and from Milan (n = 560) in Northern Italy. For participants in Bari, the study was approved by the local Ethical Committee of the Department of Education, Psychology, Communication at the University of Bari “Aldo Moro” (Ethics reference code: ET-21-06; ET-21-17), while for participants in Milan the study received approval by the Ethics Committee of the Department of Neurological and Behavioral Sciences at the University of Pavia (111/22). Participants from Bari consisted mainly of Bachelor’s and Master’s students in Psychology, Educational Sciences, History, and Philosophy, and questionnaires were administered via online surveys from September 2022 to April 2023 as part of multiple projects including eBMRQ. In contrast, the Milan sample consisted of individuals who attended a conference-concert, followed by a scientific experiment (not directly related to the eBMRQ research). The following day, i.e., from their houses, once the event finished, on a voluntary basis, they completed the eBMRQ. An informed written consent was obtained from the participants prior to the experiment. There was no time limit imposed on completion of the questionnaire. Prior to the questionnaire, participants were asked to provide demographic information, as well as to classify themselves based on their musical background as non-musicians, amateurs, or musicians. However, information on musicianship was collected from only 826 participants. This is because the data were gathered from various projects that included the Italian eBMRQ, and a few of these projects did not require specific information on music training. As a result, this information is missing for 186 participants. Among the 826 participants, 506 self-identified as non-musicians, 257 as amateurs, and 63 as musicians. Table 1 describes information on age, gender, and musicianship as a function of location (i.e., Milan vs. Bari).

In order to assess the convenience of the sample size to compute factor analysis, we used the Seneca estimate method (Lorenzo-Seva & Ferrando, 2024). This method estimates the sample size that is needed in order to reproduce the expected population correlation matrix given a precision threshold. We used as threshold a value of .003 for the root mean square of residuals (RMSR) between (1) the observed correlation matrix in the sample and (2) the population correlation matrix. If the sample at hand is close to the value proposed by Seneca Estimate, then the size of the sample is expected to be adequate for the precision proposed.

Data analysis

Unrestricted confirmatory factor analysis

The examination of the item scores of eBMRQ showed that the distributions were generally skewed. Hence, the item scores were treated as ordered-categorical variables, and the factor analysis based on the polychoric inter-item correlations was the model chosen to fit the data. This model is an alternative parameterization of the multidimensional IRT graded response model (Ferrando & Lorenzo-Seva, 2013). In order to assess the adequacy of matrix correlation to be factor analyzed, Kaiser-Meyer-Olkin (KMO) test for sampling adequacy was computed (Kaiser & Rice, 1974). Indices normed-MSA were also inspected for the items in order to decide if some item was not sharing enough communality with the whole set of items: values of normed-MSA below .50 suggest that the item does not measure the same domain as the remaining items in the pool, and so that it should be removed (Lorenzo-Seva & Ferrando, 2021). In order to compute the confirmatory factor analysis, we opted for unrestricted confirmatory factor analysis based on the Objectively Refined Target Matrix method (Lorenzo-Seva & Ferrando, 2020). The unrestricted approach is based on (1) defining the model parameters that are expected to be zero in the population, and (2) allowing these model parameters to fluctuate around zero when fitting the sample model. Nowadays, the unrestricted approach seems more suitable to assess factor models than the classical restricted approach (see for example Nájera, Abad & Sorrel, 2023). We expected a factor model in which each item was related to one of the six facets that conform eBMRQ, and that the six facets are correlated so that an overall score for each individual can be interpreted.

Table 1 Age, gender, and musicianship of the participants as a function of the location (Milan & Bari).

	Demographics	Milan sample	Bari sample	
N		556	456	
Gender	Male	229	33	
Female	320	417	
Non-binary/unidentified	7	6	
Age	Mean	46.38	20.97	
Standard deviation	14.84	3.0	
Range	18/86	18/46	
Musicianship	Non-musicians	287	219	
Amateurs	223	34	
Musicians	45	18	

The factor analysis solution was fitted by using Robust FA based on the unweighted least squares (ULS) criterion, and the chi square statistic was scaled using LOSEFER (Lorenzo-Seva & Ferrando, 2023). Currently it is usual to scale the chi square statistic using the mean and variance of the distribution of the statistics, LOSEFER goes a step further and adds the skewness and the kurtosis of the distribution of the statistic. Even if this approach is a computing-intensive task, the value of the scaled chi square is more accurate. In order to assess the goodness-of-fit of the factor model, the following indices were inspected: root mean square error of approximation (RMSEA; values between 0.050 and 0.080 are considered as fair), Comparative Fit Index (CFI; values larger than .990 are considered excellent), Goodness of Fit Index (GFI; values larger than .990 have been recommended to represent good fit), and root mean square of residuals (RMSR), with a threshold value adapted to each dataset analyzed (Harman, 1976).

The target matrix proposed was defined based on the facets in which each item was expected to load: the expected salient loading value in the corresponding facet was set as a free value in the target matrix, while the other values in the target were set to zero (i.e., we expected the corresponding loading values to show values close to zero in the loading matrix). In order to allow the analysis to refine the proposed target, we used Objectively Refined Target Matrix (RETAM) strategy (see Lorenzo-Seva & Ferrando, 2020, for further details). As a large number of values in the target were set to be close to zero, the RETAM strategy used was “To allow the model to be more complex”. It means that RETAM was allowed to add free values in the target matrix if necessary. In addition, we also defined the inter-factor correlation values to be free.

While eBMRQ is expected to be composed of six facets, the overall score in the questionnaire is also computed and interpreted. For this reason, we also assessed whether a unidimensional factor model can be expected to be reasonable in the population. In order to assess essential unidimensionality, unidimensional congruence (UNICO), explained common variance (ECV), and mean of item residual absolute loadings (MIREAL) were inspected (Ferrando & Lorenzo-Seva, 2018). Values of UNICO larger than .95, ECV larger than .85 and MIREAL lower than .30 suggest that data can be treated as essentially unidimensional. Finally, as the indices to assess essential unidimensionality suggested that a single dimension could be interpreted, the unidimensional factor solution was also inspected. Unrestricted confirmatory factor analysis was computed using FACTOR software (Lorenzo-Seva & Ferrando, 2006).

Quality of factor solution and psychometric proprieties

In order to estimate factor reliabilities of factor scores, we computed ORION reliability index of factor scores: values above .80 can be considered as acceptable. The quality of the factor solution was inspected using the construct replicability, and the quality of factor scores estimates were assessed with indices H, and sensitivity ratio (SR-index) (Ferrando & Lorenzo-Seva, 2018). The H index evaluates how well a set of items represents a common factor. It is bounded between 0 and 1 and approaches unity as the magnitude of the factor loadings and/or the number of items increase. High H values (>.80) suggest a well-defined latent variable. The SR-index can be interpreted as the number of different factor levels than can be differentiated on the basis of the factor score estimates. If factor scores are to be used for individual assessment, marginal reliabilities above .80, and SR above 2 are recommended.

Effect of age, gender, and musicianship

As additional exploratory analyses, we aimed to investigate the associations between eBMRQ, including subscale scores, gender (females/males) and musicianship (non-musicians, amateurs, and musicians). To this extent, we performed Kruskal–Wallis analysis of variance (ANOVA) due to the non-normal distribution of the data. As for the relationships between the Italian eBMRQ (the six factors and overall scale) and age, we computed Spearman Analyses using age as continuous variable. In addition, considering the wide age range of our sample (18–86 years), we grouped participants based on their age into young (18–30 years, n = 536), adults (31–60 years, n = 378), and older adults (61 years and older, n = 98). This allowed us to perform a Kruskal–Wallis ANOVA to enable a cross-sectional analysis of eBMRQ scores across the lifespan. For multiple comparisons, Dwass-Steel-Critchlow-Fligner (DSCF) post-hoc tests were used to explore differences between groups. All analyses concerning eBMRQ subscales were conducted considering the factor scores obtained using FACTOR software (Lorenzo-Seva & Ferrando, 2006), due to the small number of items in each factor. All factor analyses were computed using FACTOR, whereas analyses on age, gender, and musical expertise were conducted using Jamovi 2.3.26. Figures were generated through Jamovi 2.3.26.

Results

Sample and correlation matrix assessment

Seneca Estimate proposed that the size of the sample should be 1,129. While our sample was not that large, the difference was small enough to accept that the difference between the sample and the population reproduced correlation matrices must be close to RMSR = .003.

The inter-item polychoric correlation matrix had good sample KMO = .895 (bootstrap 95% confidence interval .859 and .927). Normed-MSA for items ranged between .736 and .967. The conclusion is that correlation matrix was suitable to be analyzed using factor analysis, and that all the items contributed substantially to the common variance.

Unrestricted confirmatory factor analysis

The factor analysis solution reached acceptable goodness-of-fit levels: RMSEA = .037, CFI = .995, and GFI = .995. In addition, RMSR = .023 was lower than Kelley’s criterion (0.032). The value of these goodness-of-fit indices allows us to conclude that the model can be expected to be observed in the general population. RETAM introduced four changes in the proposed target matrix. The changes were related to items 2 (MS-) and 20 (MS+): in both items an extra value was set as a free value corresponding to the facet SR. In addition, the two target values related to item 4 (MR) were set as free values: they were the target values related to facets SR and MS.

The loading matrix, shown in Table 2, was inspected in order to assess whether the patterns of items corresponded to the expected solution. While the overall pattern of loadings was as expected, some items have a complex pattern. The only two items that showed a large loading value in an unexpected facet were items 4 (MR) and 19 (SR).

Table 2 Factor solutions of multidimensional and unidimensional factor analysis.

Items	SR	MS	EE	MR	SM	AM	eBMRQ	
1. SR	.283	.231	.160	.077	.143	−.188	.511	
2. MS-	.372	−.901	.014	−.095	−.006	.071	−.523	
3. EE	−.043	−.049	1.032	−.022	−.019	−.152	.543	
4. MR	−.652	1.012	.064	.236	.046	−.013	.666	
5. SM-	.065	−.020	.003	.080	−.856	.061	−.347	
6. AM	.043	.283	.160	−.011	−.049	.440	.750	
7. SR	.539	.172	.044	.180	.135	−.171	.639	
8. MS	.234	.562	.013	−.092	−.088	−.006	.519	
9. EE	−.121	.102	1.003	−.036	.004	−.094	.649	
10. MR	−.052	−.001	−.016	.965	−.002	−.073	.676	
11. SM	.060	−.073	−.060	.039	.979	−.039	.488	
12. AM	−.091	.039	−.025	.062	.035	.818	.737	
13. MS	−.044	.720	−.194	−.035	.067	.149	.577	
14. EE	.075	−.172	.555	.021	.083	.212	.586	
15. SR	.642	−.038	.023	−.065	.015	−.051	.339	
16. MR	.041	−.110	−.066	1.063	−.043	.023	.742	
17. SM	.017	.036	.080	.146	.464	−.039	.472	
18. AM	−.159	.068	−.021	−.028	−.055	.986	.726	
19. SR	.142	.174	.039	.081	.228	.061	.544	
20. MS	.473	.346	−.055	−.088	−.174	.073	.462	
21. EE	.138	−.113	.473	.057	−.063	.314	.652	
22. MR	−.004	.203	.105	.479	−.067	.182	.780	
23. SM	−.008	.039	.033	−.141	.756	.178	.529	
24. AM	.197	−.234	−.020	.028	.098	.743	.649	
Notes.

The bold face highlights the loading value that was expected to be the largest one for each item. The loading value in each expected factor are printed in bold face.

The dimensions were related to a reasonable number of salient loadings (i.e., loadings with absolute value larger than .25): 3 (facet MR), 4 (facets EE and SM), 5 (facets MS and AM) and 6 (facet SR). The facet with the lowest ORION reliability was SR (value of .806), while the other facets showed reliability that ranged from .891 to .959. These outcomes support the hypothesis of a six-dimensional solution. The values of the inter-factor correlation matrix ranged between .245 and .769 (see Table 3). In addition, all the facets showed at least three correlations larger than .4 with other facets. These large correlation values among facets suggest that a unidimensional factor solution might also be acceptable.

Table 3 Inter-factor correlation matrix among the six dimensions.

Dimensions	SR	MS	EE	MR	SM	
MS	.769					
EE	.476	.530				
MR	.420	.678	.592			
SM	.245	.343	.400	.437		
AM	.659	.735	.600	.632	.431	

To assess the essential unidimensionality, the values of the indices to assess essential unidimensionality were UNICO = .901 (bootstrap 95% confidence interval .882 and .919), ECV = .799 (bootstrap 95% confidence interval .781 and .816), and MIREAL = .226 (bootstrap 95% confidence interval .205 and .248). These outcomes suggested that a single dimension could also be acceptable for the questionnaire. The factor analysis with a single factor reported the following goodness-of-fit indices: RMSEA = .085, CFI = .952, GFI = .937, RMSR = .1058. While the fit worsened, it was still acceptable. The absolute loading values in the single factor ranged from .339 to .780. ORION reliability for the factor was .939.

In sum, the unrestricted confirmatory analysis indicated that the six-factor model was acceptable for the eBMRQ, and that a single factor would also be informative in order to compute an overall score in the questionnaire.

Quality of factor solution and psychometric proprieties

The indices to assess the quality of the factor solution are showed in Table 4. As previously described in Cardona et al. (2022), H-Latent evaluates how well the factor can be identified by the continuous latent response variables underlying the observed item scores (i.e., the factor scores estimates), while H-Observed assesses how well it can be identified from the observed item scores (i.e., the raw total of participants’ responses to the items). Since the H-Latent values are systematically higher than the corresponding H-Observed values, factor scores should be computed rather than simply adding up the item response. Therefore, instead of simply adding up the responses to items, it is better to compute factor scores to capture the true underlying traits or abilities measured by the test. This approach takes into account the relationships between items and the latent variables they are intended to measure. In the supplementary materials, we provide researchers with an excel file that computes factor scores from participants’ responses to the items and raw data are freely available at the Open Science Framework (OSF) (https://osf.io/ka7xc/).

Table 4 Indices to assess the quality of the factor solution.

Index	SR	MS	EE	MR	SM	AM	eBMRQ	
H-Latent	.806	.891	.917	.959	.934	.914	.939	
H-Observed	.767	.845	.712	.841	.837	.871	.902	
SR-index	2.038	2.860	3.799	4.854	3.762	3.255	3.921	

The value of the indices indicate that both the six dimensions and the overall dimension exhibit an acceptable quality. The facet with the lowest quality is SR. Overall, the indices for the unidimensional solution are higher than those for the six dimensions. This suggests that the factor score for the overall questionnaire is more reliable and informative than the scores for the individual facets composing the test, as also suggested in Cardona et al. (2022).

For the following sections, we computed eBMRQ factor scores that are the ones related to the unidimensional factor model (i.e., the overall factor score), and the factor scores related to the six dimensions (i.e., Musical Seeking, Emotion Evocation, Mood Regulation, Social Reward, Sensory-Motor, and Absorption in Music). Please note that, as two items did not load in the expected dimensions, the most informative outcome and comparable to the original study is the one related to eBMRQ scores.

Relationship between eBMRQ scores and gender

To assess the putative gender effect on eBMRQ scores we computed a Kruskal–Wallis ANOVA due to abnormal distribution of the data. 13 participants chose not to specify their gender or identify as non-binary and were thus excluded from the analysis. Results revealed that females scored significantly higher than males in the eBMRQ overall scale (H(1) = 13.96, p < .001, ɛ2 = 0.013) (see Fig. 1 for a graphical depiction of the results). Specifically, females showed significant higher values in the Emotion Evocation (H(1) = 9.10, p = .003, ɛ2 = 0.009), Mood Regulation (H(1) = 20.91, p < .001, ɛ2 = 0.020), Sensory-Motor (H(1) = 120.17, p < .001, ɛ2), and Absorption in Music scale (H(1) = 5.09, p = .024, ɛ2 = 0.005). In contrast, males’ scores on Social Reward scale were significantly higher than females (H(1) = 13.34, p < .001, ɛ2 = 0.013). No gender difference arose in Musical Seeking scale.

Figure 1 Box plot illustrating the distribution and means of eBMRQ scores in females and males.

Musicianship

To evaluate the effect of musicianship on eBMRQ scores we asked participants to classify themselves as non-musicians, amateurs, or musicians based on their musical background. By computing Kruskal-Wallis ANOVA we found a significant difference between the groups in the eBMRQ overall scale (H(2) = 48.06, p < .001, ɛ2 = 0.058) (see Fig. 2 for a graphical depiction of the results). Specifically, DSCF pairwise comparisons suggest that musicians and amateurs’ overall scores were significantly higher than non-musicians, Z = 5.87, p < .001, and Z = 8.82, p < .001, respectively. The effect of musicianship arose in all eBMRQ scales except for Mood Regulation and Sensory-Motor (H(2) = 4.87, p = .088, and (H(2) = 5.48, p = .064, respectively). All results are summarized in Tables 5 and 6.

Figure 2 Box plot illustrating the distribution and means of eBMRQ scores in non-musicians, amateurs, and musicians.

Age

As data were not normally distributed, the effect of age was first explored by conducting a Spearman rank correlation coefficient analysis between the age of the participants and eBMRQ overall scores. This analysis indicates that eBMRQ overall scores are negatively correlated with age (Spearman’s rho = −0.117, p < .001). Significant negative associations with age were specifically observed in Mood Regulation (Spearman’s rho = −0.157, p < .001), Sensory-Motor (Spearman’s rho = −0.262, p < .001), and absorption scales (Spearman’s rho = −0.082, p = .009), whereas for Social Reward scale the correlation with age was significantly positive (Spearman’s rho = 0.131, p < .001). To allow for a deeper analysis, these significant associations were further explored using Kruskal-Wallis ANOVA taking into account the participants’ age group (young, adults or older adults) (see Fig. 3 for a graphical depiction of the results). The results confirmed the significant effect of age on eBMRQ overall scores (H(2) = 16.7, p < .001, ɛ2 = .0165), Social Reward (H(2) = 20.95, p < .001, ɛ2 = .0207), Mood Regulation (H(2) = 24.09, p < .001, ɛ2 = .0238), Sensory-Motor (H(2) = 65.81, p < .001, ɛ2 = .0650), Emotion Evocation scale (H(2) = 6.04, p .049, ɛ2 = .005, and Absorption in Music scales (H(2) = 8.60, p .014, ɛ2 = .008. DSCF pairwise comparisons revealed that eBMRQ overall scores are significantly different between young and older adults (Z = −5.74, p < .001, r = −.23), as well as between adults and older adults’ groups (Z = −4.06, p = .012, r = −.19). The only scale that significantly differed in each age bracket was the Sensory-Motor: young vs. adults (Z = −8.94, p < .001, r = −.30), young vs. older adults (Z = −9.13, p < .001, r = −.36), and adults vs. older adults (Z = −3.93, p = .015, r = −.18). All results are included in Tables 7 & 8.

Table 5 Kruskal–Wallis ANOVA as a function of musicianship.

	H2	df	P value	
eBMRQ (overall sum)	48.06	2	<.001	
Social reward	152.09	2	<.001	
Musical seeking	76.34	2	<.001	
Emotional evocation	37.96	2	<.001	
Mood regulation	4.87	2	0.088	
Sensory-motor	5.48	2	0.064	
Absorption in music	55.11	2	<.001	

Table 6 DSCF pairwise comparisons as a function of musicianship.

eBMRQ and subscales scores	Musicianship	Wilcoxon Z	P value	
eBMRQ (overall sum)	Non-musicians vs. Amateurs	8.82	<.001	
Non-musicians vs. Musicians	5.87	<.001	
Amateurs vs. Musicians	1.19	0.677	
Social reward	Non-musicians vs. Amateurs	14.50	<.001	
Non-musicians vs. Musicians	12.01	<.001	
Amateurs vs. Musicians	5.40	<.001	
Musical seeking	Non-musicians vs. Amateurs	11.01	<.001	
Non-musicians vs. Musicians	7.59	<.001	
Amateurs vs. Musicians	1.68	0.459	
Emotional evocation	Non-musicians vs. Amateurs	7.74	<.001	
Non-musicians vs. Musicians	5.33	<.001	
Amateurs vs. Musicians	1.67	0.464	
Mood regulation	Non-musicians vs. Amateurs	2.874	0.105	
Non-musicians vs. Musicians	−0.694	0.876	
Amateurs vs. Musicians	−2.144	0.283	
Sensory-motor	Non-musicians vs. Amateurs	−1.88	0.377	
Non-musicians vs. Musicians	−3.08	0.075	
Amateurs vs. Musicians	−1.73	0.437	
Absorption in music	Non-musicians vs. Amateurs	8.38	<.001	
Non-musicians vs. Musicians	7.67	<.001	
Amateurs vs. Musicians	3.61	0.029	

Figure 3 Box plot illustrating the distribution and means of eBMRQ scores in young, adults, and older adults.

Discussion

The experience of pleasure in music has long been recognized as a crucial aspect of human engagement with this art form. The rewarding experience of music is, however, considerably subjective and diversified (e.g., Garrido & Schubert, 2011; Martínez-Molina et al., 2019; Zatorre, 2015). Validating a tool that can accurately and reliably measure such individual differences is thus crucial. The extended Barcelona Music Reward Questionnaire (eBMRQ) is a recent version developed by Cardona et al. (2022) which adds to the five scales of the original BMRQ by Mas-Herrero et al. (2013) (i.e., Sensory-Motor, Musical Seeking, Mood Regulation, Social Reward, and Emotional Evocation) a further scale assessing absorption in music, a crucial component indicating the profound emotional involvement and connection experienced when exposed to music (Hall, Schubert & Wilson, 2016; Kreutz et al., 2008).

Whereas an Italian version of the BMRQ has recently been published (Mannino et al., 2024), an Italian validation of the eBMRQ was still lacking. With this study, we developed and validated an Italian version of the extended Barcelona Music Reward Questionnaire (eBMRQ) across a large sample of Italian participants (n = 1,012). We also provided evidence on the effect of age, gender, and musicianship on individual music reward sensitivity, as indexed by eBMRQ and its relative subscales’ scores. For the validation process, we conducted an unrestricted confirmatory factor analysis based on Objectively Refined Target Matrix method (Lorenzo-Seva & Ferrando, 2020). The unrestricted approach is deemed to be more suited for evaluating factor models than the conventional restricted approach (Nájera, Abad & Sorrel, 2023). As with the original English version, the overall Italian version of the eBMRQ showed acceptable psychometric properties, indicating that the Italian version of the eBMRQ is a reliable tool to assess sensitivity to music reward. In addition, correlations between the various factors that emerged are strong, with values ranging from .245 to .769. Importantly, while the original version of the test proposes a factor model based on six facets (each facet defined by four items), the outcomes observed in the Italian version of the test suggest that not all the facets are equally defined as in the English version of the test. In particular, two items seem to load in different facets than expected. In addition, the assessment of the essential unidimensionality suggests that the unidimensional solution is a plausible factor model for the Italy population. We would suggest to Italian applied researchers to be careful when interpreting the factor scores of the six facets because the sub-dimensions in Italian might be less valid than in the original test. On the other hand, we would also advise researchers to base their assessments in the overall factor score (i.e., the one obtained that considering the test as a unidimensional test). The two items which loaded into unexpected factors were 19 and 4. Item 19 (“At a concert I feel connected to the performers and the audience”) originally intended for the Social Reward factor, mainly loaded into the Sensory-Motor factor. One possible interpretation is that participants assessed their “connection” with performers and the audience mainly in terms of physical involvement and synchronization. The unexpected loading of item 4 (“Music keeps me company when I’m alone”) into Musical Seeking rather than Mood Regulation could instead potentially be attributed to the characteristics of nearly half of our sample (n = 560), who completed the questionnaire the day after participating to a conference-concert they had willingly attended. It is plausible that these participants associated the sense of being less alone through music, and more broadly, the ability to feel better with music, with the need to actively seek music in their daily lives (i.e., musical seeking). On this line, a recent study conducted in the context of music and COVID suggested that engaging in activities such as listening to new music and attending virtual concerts (i.e., musical seeking activities) was associated with the modulation of positive feelings and served as a way to socialize, respectively (i.e., mood regulation and feeling less alone) (Ferreri et al., 2021). In general, the fact that these two items show the main loading value in a different factor than in previous factor analyses is not an issue for this work. Indeed, we observed that the test is essentially unidimensional, with highly correlated factors representing facets of the main factor. Thus, these items still contribute substantially to the main factor.

Table 7 Kruskal–Wallis ANOVA as a function of age group.

	H2	df	P value	
eBMRQ (overall sum)	16.71	2	<.001	
Social reward	20.95	2	<.001	
Musical seeking	2.72	2	0.257	
Emotional evocation	6.04	2	0.049	
Mood regulation	24.09	2	<.001	
Sensory-motor	65.81	2	<.001	
Absorption in music	8.60	2	0.014	

Table 8 DSCF pairwise comparisons as a function of age group.

eBMRQ and subscales scores	Age group	Wilcoxon Z	P value	
eBMRQ (overall sum)	Young vs. Adults	−2.35	0.219	
Young vs. Older adults	−5.74	<.001	
Adults vs. Older adults	−4.06	0.012	
Social reward	Young vs. Adults	6.075	<.001	
Young vs. Older adults	3.693	0.024	
Adults vs. Older adults	−0.267	0.981	
Musical seeking	Young vs. Adults	−0.212	0.988	
Young vs. Older adults	−2.333	0.225	
Adults vs. Older adults	−2.052	0.315	
Emotional evocation	Young vs.. Adults	2.13	0.289	
Young vs. Older adults	−1.95	0.352	
Adults vs. Older adults	−3.46	0.038	
Mood regulation	Young vs. Adults	−5.16	<.001	
Young vs. Older adults	−5.77	<.001	
Adults vs. Older adults	−2.65	0.147	
Sensory-motor	Young vs. Adults	−8.94	<.001	
Young vs. Older adults	−9.13	<.001	
Adults vs. Older adults	−3.93	0.015	
Absorption in music	Young vs. Adults	−1.23	0.661	
Young vs. Older adults	−4.17	0.009	
Adults vs. Older adults	−3.24	0.057	

The results indicating that females’ eBMRQ scores are significantly higher than males align with several BMRQ studies which pointed out a similar tendency (Fuentes-Sánchez et al., 2023; Hernández et al., 2019; Mas-Herrero et al., 2013). Consistent with the original eBMRQ study (Cardona et al., 2022), Musical Seeking scale scores did not show significant differences based on the gender. However, in contrast to the latter study, we found that scores in Absorption in Music scale were significantly higher among females than males. In general, as our sample consisted of significantly more females than males (n = 737 and 262, respectively), the interpretation of gender results should be cautious as more gender-balanced research is needed.

An additional aim of our study was to explore the relationship between age and music reward sensitivity across the Italian population. Consistently with prior evidence (Belfi et al., 2022; Cardona et al., 2022), we observed a negative correlation between age and eBMRQ overall scores Specifically, the effect of age is significantly negative in the overall, Sensory-Motor, Mood Regulation, and Absorption in Music scales, whereas it is significantly positive in the Social Reward scale. This latter result is not consistent with previous studies, although the relationship between age and the Social Reward scale has previously yielded conflicting results. For instance, Mas-Herrero et al. (2013) did not report significant associations between Social Reward scores and age, whereas Cardona et al. (2022) found a negative correlation. The reason for these inconsistencies may be linked to the highly activity-specific nature of the items comprising the scale, such as “I like to sing or play an instrument with other people” or “At a concert I feel connected to the performers and the audience”.

The non-significant effect of age on Emotion Evocation scale is in line with Cardona et al. (2022). However, contrary to the latter study, we found a negative correlation between age and Absorption in Music scale, although this significant effect is marginal (p = .049). These discordant results might be due to the fact that those in our sample who scored lower on the Absorption in Music were older adults (i.e., 61 years and older, n = 98), and in Cardona’s study only 20 out of 920 individuals fell in this age category. A comprehensive interpretation of the results might be that the decrease in musical reward with advancing age may primarily arise from reduced physical engagement in music-based activities (e.g., dancing), especially considering that older adults may experience mobility issues (Lauretani et al., 2003). Conversely, musical pleasure derived from the emotional and evocative aspects of music, as depicted in the Absorption in Music and Emotion Evocation scales, tends to be less sensitive to aging. Furthermore, in our study, we did not observe any age-related effect on Musical Seeking scale, which assesses the extent of individuals’ engagement in music-related activities. This is in contrast with Cardona et al. (2022) and Mas-Herrero et al. (2013) who both reported a negative association. Once again, this incongruent evidence could stem from the fact that many of our participants (n = 560) were recruited during a public event also involving music held in Milan. Therefore, a certain number of participants of this study might have been intrinsically motivated in the participation of music-based activities, supposedly affecting their Musical Seeking scores.

Given our extensive and diverse sample in terms of age (n = 1,012, age range 18–86), we conducted an additional analysis of eBMRQ scores across different age groups (young, adults, older adults). This further analysis pointed out that the decline in musical reward sensitivity is consistent across the lifespan, but it is significant only in young vs. older adults, and adults vs. older adults comparisons. In contrast, the Sensory-Motor is the only scale in which the score significantly decreases across all three age brackets. Lastly, we wanted to explore the relationship between musicianship and eBMRQ scores. Results indicate that musicians and amateurs scored significantly higher than non-musicians in the overall eBMRQ score. The positive association between musical expertise and musical reward has been previously observed in both behavioral (Fuentes-Sánchez et al., 2023; Mas-Herrero et al., 2013), and neuroimaging studies (Alluri et al., 2015). This effect appears significant in all eBMRQ scales except for Mood Regulation and Sensory Motor. Hence, also in light of the aforementioned results, the Sensory-Motor scale appears to be more sensitive to the effect of age rather than the level of musicianship.

Limitations

This work includes some limitations which should be acknowledged. Firstly, although the age range of our sample was quite wide (18–86 years), the generalizability of our study may be limited by the fact that nearly half of the participants were university students (452 out of 1,012). Secondly, in our study musicianship was assessed based on participants’ self-identification as musicians, amateurs, or non-musicians, with musicians being only 8% of the final sample. A more balanced and in-depth analysis, based on factors such as years of musical practice and/or type of instrument played, may certainly provide more precise insights into the relationship between musical expertise and music-related pleasure. Moreover, regarding the effect of age on eBMRQ scores, it should be noted that as our analysis is cross-sectional, we cannot establish a cause-and-effect relationship or analyze behavior over a period of time. In this regard, a longitudinal study on the trajectory of sensitivity to musical pleasure during the lifespan could provide further reliable evidence. Future research should delve deeper into the underlying mechanisms driving these age-related variations in music reward sensibility, with potential implications for promoting well-being and enhancing the musical experiences of individuals across the lifespan.

Conclusions

The results we have reported suggest that the Italian version of the eBMRQ exhibits acceptable and reliable psychometric properties, making it a valid tool for assessing sensitivity to musical reward within the Italian population. Furthermore, the significant size of the sample (n = 1,012) has enabled us to provide valuable insights into the effects of individual differences such as age, gender, and musicianship on the construct of music reward sensitivity. The application of this tool in research can help delve further into the exploration of how individual differences, cultural influences, and various musical contexts shape the experience of musical pleasure. This expanded scope may potentially not only contribute to the advancement of music psychology but also provide valuable insights into the broader understanding of human enjoyment and well-being.

Supplemental Information

Supplemental Information 1 Italian version of the extended Barcelona Music Reward Questionnaire (eBMRQ)

Supplemental Information 2 Italian eBMRQ factorial scores calculator

Supplemental Information 3 STROBE checklist

Supplemental Information 4 Participants’ demographic information and raw data of eBMRQ scores

Additional Information and Declarations

Competing Interests

Author Contributions

Human Ethics

Data Availability

The authors declare there are no competing interests.

Giulio Carraturo conceived and designed the experiments, performed the experiments, analyzed the data, prepared figures and/or tables, authored or reviewed drafts of the article, and approved the final draft.

Laura Ferreri conceived and designed the experiments, performed the experiments, authored or reviewed drafts of the article, and approved the final draft.

Gemma Cardona performed the experiments, authored or reviewed drafts of the article, and approved the final draft.

Urbano Lorenzo-Seva analyzed the data, prepared figures and/or tables, authored or reviewed drafts of the article, and approved the final draft.

Antoni Rodriguez-Fornells conceived and designed the experiments, performed the experiments, authored or reviewed drafts of the article, and approved the final draft.

Elvira Brattico conceived and designed the experiments, performed the experiments, authored or reviewed drafts of the article, and approved the final draft.

The following information was supplied relating to ethical approvals (i.e., approving body and any reference numbers):

For participants in Bari, the study was approved by the local Ethical Committee of the Department of Education, Psychology, Communication at the University of Bari “Aldo Moro” (Ethics reference code: ET-21-06). For participants in Milan, the study was approved by the Ethics Committee of the Department of Neurological and Behavioral Sciences at the University of Pavia (n° 111/22).

The following information was supplied regarding data availability:

The raw data is available in the Supplemental Files.

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
