# Peer review of "The Italian version of the extended Barcelona Music Reward Questionnaire (eBMRQ): a validation study and association with age, gender, and musicianship"

_PeerJ, doi:10.7717/peerj.18403_

## Round 0.1 · original submission · Major Revisions

As you can see, the reviewers have offered constructive feedback that I believe will greatly assist you in revising your manuscript. I kindly request that you provide comprehensive responses to each comment from the reviewers.

·

Basic reporting

The basic reporting is at sufficient level although the contextualisation could be either more brief (as this is a translation of a tool) or more expanded (the literature cited is narrow, especially for mood regulation and pleasure).

Experimental design

- Participant selection could be clarified: this seems to be a fairly narrow convenience sample (uni students) complemented with a more diverse sub-sample during an event.
- I wonder why missing data needs to be imputed if there was a single missing number and 1012 participants studied.
- The definition of musical expertise seems to be quite ad-hoc which is disappointing since there are so many standard measures for this (Gold-MSI, OMSI, etc.). This is one caveat the probably impacts the anova results.
- The estimation of the sample size seems to be undefined ("We used as threshold a value of .003 for the Root Mean Square of Residuals") or at least the reader is not given any standards here.

Validity of the findings

I am pretty much familiar with conventional EFA/CFA techniques, but I have to admit that I am not familiar with some of the techniques used here (e.g. LOSEFER, Objectively Refined Target Matrix, ORION for reliability), which are all unconventional and seem to be rarely used. I am not saying they are wrong, but since this is basically testing something that we already have (BMRQ), it would make sense to use the same methods as the original development of the BMRQ and show comparable indices to other studies.
- In the ANOVA analysis, the authors reveal that the data is non-normally distributed, but this could have been communicated earlier and the implications for analytical operations considered earlier. For instance, right after following the announcement of the non-normal data, the authors carry out "Pearson's Analyses" between factor cores and age, but of course one should be using a less biased correlation method (e.g. spearmann rank) if the data indeed is non-normally distributed.
- When authors state that "All analyses were computed using FACTOR, SPSS, and Jamovi" one could be more specific (which analyses with what software, which versions).
- Why are polychoric correlations used when the data consist of likert scale ratings (1-5)?
- Division of age into three groups is not really motivated clear, how was this constructed and is it even a good idea to make continuous age data discrete since one could analyse this links without the categories. If those three are the categories, are the results with different cut-off points and so on. So this age group and musical expertise feel a bit arbitrary and they get a lot of attention but the premises need to be clarified.

Additional comments

Overall, these are small technical notes, the bulk of the study and the results seem to be communicating a very clear pattern of results with a large sample and rigorous back translation protocols that suggest that the Italian version of the eBMRQ is a valid and reliable instrument. Great work, overall!

Reviewer 2 ·

Basic reporting

The present study aimed to validate the italian version of the eBMRQ. It was administered to more than a thousand participants coming from Milan and Bari. Psychometric properties were found to be good apart from two items that do not load in the original planned factors. The authors checked also for the influence of age, gender and musical expertise, with results mostly in line with previous literature.
The manuscript is overall clear and well written, however I have some concerns regarding some methodological choices and how certain findings were addressed (see specific comments in the methods and findings sections).
Raw data is provided for review but not in the manuscript.
I would replace figures with boxplots (see comment in the findings section)
More details will be reported in the various sections below. Minor comments will be added at the end.

Experimental design

The study has a generally good design, however I believe that there is a major limitation that the authors should addressed more (maybe adding a limitation section would also be nice). Participants were selected from the north and south of Italy, to allow for "greater generalizability" (as the author state in row 150). However, by looking at the details of the populations chosen, they do not appear a great choice to allow for generalization. First, in Milan the sample was taken from a music event. I encourage the authors to describe, also, what kind of event this was (e.g., a concert, a workshop?). This means that most of these participants (if not all) are already highly interested in music, and this could over-represent the general distribution of music interest in the whole population. Moreover, as the authors clearly stated, this could have influenced why some of the items did not load in the "correct" factor. If this interpretation is true, in my opinion, it is quite problematic, as if the way people are recruited (or the fact that they are particularly active in music seeking), biases some of the responses in the items, this means that those items are not good for the general population, as different interpretation could lead to different results. Of course a certain amount of free interpretation is always possible, but if the authors are aware that this is a potential problem, they should try to rephrase the item, or remove it, or give clear suggestions on how to handle these items for those who will administer the Italian version in the future.
-The other sample consists of psychology and education students. We know how problematic for generalization is taking just psychology (and university students). I would address this also as a limitation. In general, my suggestion here would be (if not getting more data), to at least add a comparison between the two groups, first in terms of demographics (and music expertise), and then an exploratory factor analysis and reliability separated for the two samples. I am aware this could lead to an underpowered statistical analysis, but in this case it would just be needed to understand the potential cause of those two items having the wrong load.
-Music expertise is not clearly describe. The authors created three groups but the actual inclusion criteria for each group are not sufficiently described. Particularly, in rows 164-167, the authors should specify what "little", "some" and "significant" level of musical training/experience mean. For examples, what are the years of training/expertise used as a cutoff for the three categories? How was formal and informal expertise assessed (which questions?). Were the hours of practice assessed? I think this categorization, without these details, appears to be too approximate for the purpose of the study. Also, it would be nice to have these details (row answers to these questions) included in the dataset too, instead of the simple labels to categorize the groups.
Moreover, the authors state that just a part of the sample was asked about music training. Can they be more precise in terms of which part ? (for example only in Milan? or only in Bari?).
- In my opinion, maybe it is better to just have two groups, given the large difference in terms of size, and merge amateur and expert musicians together.
-It is also not clear to me if the participants in Milan were all recruited at the same event (did they fill in the surveys via their smartphones directly at the event site?), as the data collection apparently went on for several months (maybe only in Bari?)
- Concerning the translation of the questionnaire. It is not clear which is the original version of the questionnaire. Is it the English one? The authors refer to the Spanish version too. Are the English and the Spanish version both validated and equal in terms of psychometric properties? Maybe in the introduction, existing versions could be mentioned.

Validity of the findings

-Data is provided, however not linked in the manuscript. I encourage the authors to provide a link for the data in the manuscript.
-Missing data. The authors state that they used multiple imputation. I think it is important to say more about how the missing values were treated (e.g., MAR) and also, how many missing values there were. I read a sentence at rows 256-257 where it seems that there is a missing value only? Does this mean that one item had no response? I think this is not clear. If it was only one incomplete answer, then I would simply remove the subject, I do not think multiple imputation here is needed. Unless the data missing is more, and in this case it would be important to know how many, and whether they were systematic (e.g., always at the end of the questionnaire because people interrupted it).
-I had some troubles following the statistical analysis (for the validation of the questionnaires), as I am not familiar with these specific statistics. They seem to refer always to the works of one of the authors. I have not the competence to say whether these methods are as good (or even better) than the more traditional ones, but I would encourage the authors to describe, perhaps at the beginning of the data analysis section, why these specific methods are used, how are they comparable to the most traditional ones, and what is their advantage. I read a sentence about this in the discussion (rows 382-383), however I would prefer to elaborate more on this in the method section. Also, some specific details are not clear to me. For example, for reliability the authors used the "ORION" index, but this does not seem to be a common measure of reliability. It would be nice to read how this is interpreted and whether it is similar, for example, to Cronbach alpha, in the way it is computed.
-The fact that two items have unexpected loadings is a problem. The authors should either remove them, or try to rephrase them and test them again, or give detailed instructions on how to deal with this. If somebody wants to use this scale and they are interested in one of the factors specifically, then can the factor be considered valid anyway even with an item that does not specifically load on that factor? or should this be excluded for the single factor score? If this has to be excluded, is the questionnaire still reliable, as well as the factor?
-At row 283 the authors mention that the inter-factor correlations ranged between .245 and .769. I think it would be better to show the actual matrix with the full correlations, because reporting just the range might "hide" the fact that most of factors have a quite small correlation, even if later it is written that all of them had at least three that were higher than .40. This seems a bit vague in my opinion, for transparency better to report everything. I am also lost here, on which index is used for these correlations. About this point, also, I suggest that the authors use Spearman correlations after as for the non-normal distribution of their data they always used non-parametrical tests.
-Overall what I miss concerning the findings on the validation of the Italian version of the questionnaire, is a direct comparison with the original version (important to define also if this is the Spanish or the English one). Is it worse/better/equal? This would be important to include in the results and to discuss too.
-Concerning the analysis on demographic variables and music expertise, I suggest that the authors report the effect sizes for the comparisons. I suggest displaying the results with boxplots instead of barplots, as the full data distribution can be observed more clearly (and means can be added too).
-I am a bit insecure about the analysis on age categories. To me this seems not necessary. The correlations already show a trend (around .10, so very small correlations), and it is not straightforward to understand why the authors needed to take a look at this also by creating three different groups. Moreover, the group of "older people" include individuals from 51 years old on, but the conventional categorization of older adults start at 65 (or, in some cases at 60). I would rather keep only the correlations but I would be careful in the interpretations, as yes, they are significant, but this is likely due to the large sample size, however the size, which is what is really informative, is very small, so this should be taken into account when also comparing the current findings to previous ones.

Additional comments

Specific (minor) comments.
Some of the sentences in the introduction are a bit vague, I would suggest to add more examples or be more specific, e.g..,
- rows 61-62 "musical experiences" (examples?)
-rows 75-76 this somehow seems an explanation of the "more complex" reported just above, however I think it would be nice to read a bit more about this. Are these activations only found in music related pleasure in comparison to other stimuli? what could be the role of attention/memory? I think because music pleasure is the subject of the questionnaire, it would be nice to elaborate a bit more on the mechanisms behind.
-row 84 "factors may influence" maybe add the direction in which this happens (e.g., positively?)
-row 107 "musicians score higher than nonmusicians" in all scales? I think this would be nice to know.
-row 111 an on. Absorption. I would suggest to explain how absorption is different from the factor of Emotion Evocation, as it is described as a immersive and also emotional experience.
-rows 119-120. I would suggest to add a few examples here.
-rows 137-138 I would add a citation for the ORION index here, and perhaps how to interpret it (or state that more details are described later).
row 267. "in the population" in the general population?
rows 281-282. How are these values supporting the six-dimensional division? Is it not factor analysis that support this and not reliability?
row 290. "acceptable for the dataset" For the questionnaire maybe? Not clear here.
row 305-306. Is this the standard way users should score the questionnaire? If so, I suggest to indicate it more clearly.
row 337 (and elsewhere). Sometimes p-values have the 0 before the decimal point.
row 347. "declines as age increases" Here and there there are sentences that implicitly refer to a causal relationship. Even though the authors are aware of the correlational nature of their study and they mention this, in the choice of language a causal implication sometimes appears.
row 385. Construct validity. Not clear how this was assessed here. Maybe it is sufficient that in the original version this was assessed, but here I do not see clearly how, e.g., convergent and discriminant validity were assessed.
rows 400-404. This feels a bit weak as an explanation, either I suggest to make it clearer, or just remove it.
Table 3. "modd" instead of "mood".

---

## Round 0.2 · Major Revisions

Thank you for addressing the reviewers' concerns. Reviewer 2 has questioned several of your responses regarding item loadings compared to the original version etc.

·

Basic reporting

The authors have a done thorough job in providing the missing details and have patiently explained and expanded the issues raised.

Experimental design

Nothing extra to report here, the authors have addressed the issues raised and fixed the minor omissions.

Validity of the findings

All the issues have been addressed and the choices about the factor interpretations and age groupings have been explained in a satisfactory way.

Additional comments

I thank the authors for their diligent and valuable work on this contribution music scholarship and it is great to see careful and systematic validation and expansion of the tools to other languages and cultural contexts.

Reviewer 2 ·

Basic reporting

I have no comments in this section, everything was addressed.

Experimental design

AIMS AND SCOPE: I find still problematic that the authors did not investigate further, nor discussed properly, the problem of the two items loading in different factors from the original version. They say that investigating this and also comparing this with the original version is out of scope for the present paper. Maybe I am wrong, but to me validating a questionnaire cannot ignore the comparison with the original language. This cannot be another paper in my opinion. Otherwise this is a simple translation with a general reliability index, not a full validation. I understand that this would force the authors to propose the uni-dimensionality here, but in my opinion this is important, otherwise I cannot know as a potential user of the questionnaire whether the Italian version is what I want to use depending on my goals. It does not have to extend too much the paper, but at least in the discussion they should expand more on the fact that the sub-dimensions in Italian might be less valid (if they do not want to explore this further as suggested in my first comments) and that users wanting to assess specific sub-dimension should further validate or use different questionnaires.
I still miss which is the original version language, this was provided as a response to my comment but not in the text.

PARTICIPANTS: if the sample in Bari (I guess, but not specified) comes from different projects, there should be more ethics approvals than just one for Bari? Moreover, I would suggest to add in the table the amount of participants for which there is no musicianship status (in other words, how many NAs). Concerning the question for musicianship status, in the manuscript I can still read the description of the categories. But if the participants self-reported which category the felt belonging to, it is not clear how you can have this category description, unless this was the description given to participants? If so, please specify this, or remove the description, simply state that they selected one of these three categories (without description).
The authors mentioned that there were not just psychology students, but also others from another faculty. Now I only read that the students were mainly from educational sciences, history and philosophy. Psychology students are not mentioned anymore. This is a bit strange, please clarify or mention psychology students again.

ANALYSIS: I find the explanation of the statistical choices clearer, but I think the manuscript would benefit of the addition of interpretation guidelines of the ORION index (values cutoffs for acceptable, good, excellent reliability).


RESULTS-AGE: I find the response of the author not really convincing, concerning the decision to keep the different groups of age, especially for the categorization of older adults, which cannot simply ignore that participants between 50-60 years old are not considered conventionally older adults. One study that used this in the past is not sufficient to justify why here this was decided against the convention. In my opinion the correlations are sufficient, if not, at least adopt the conventional division of age groups.

Validity of the findings

The only thing I still believe being problematic, as mentioned in the AIMS and SCOPE above, is that the authors do not expand the validation to a comparison with the original versions. If this requires to propose a one-dimension construct, then it should be done, at least discussed briefly, it does not have to become the core of the manuscript but it cannot be ignored that some items might not be good for the sub-dimensions computation. Especially because the authors here run analyses separate for the sub-dimensions, and for future studies wanting to do the same (e.g., separate correlational analyses)
Construct validity: based on my previous comment, that was maybe misunderstood,at line 474-475 the authors have changed construct validity to convergent and discriminant validity. The sentence is now: "the overall Italian version of the eBMRQ showed acceptable psychometric properties, and acceptable convergent and discriminant validity," My question was, how is construct validity tested in this version, was convergent and discriminant validity tested? it appears to me not, so please verify this is correct and in case remove or add these validation in the methods/results section. I do not think it is a problem if construct validity was tested in the original version, but this appears to me as a wrong statement as I do not find this info in the present manuscript.

Additional comments

Minor: line 249-250: "Even if this approach is a computing-intensive task, the value of the scales chi square is more accurate".
i find this sentence not clear. Scales should be "scaled" maybe? Can you provide a reference to support this?

Line 343: This was the table with the Spearman correlations I suggested to create, however maybe there was a misunderstanding and I still see the same sentence as before. Please add full correlation matrix for this.

Line 397: effect size missing.

---

## Round 0.3 · accepted · Accept

Thank you for addressing all the concerns raised.

Reviewer 2 ·

Basic reporting

I think that the authors did a good job in addressing my comments. I think the manuscript deserves to be published in its present form.

Experimental design

All comments addressed

Validity of the findings

All comments addressed